# Disorders in brassinosteroids signal transduction triggers the profound molecular alterations in the crown tissue of barley under drought

Anetta Kuczyńska[1], Martyna Michałek[1], Piotr Ogrodowicz[1], Michał Kempa[1], Paweł Krajewski[1], Vladimiro Cardenia[2], Maria Teresa Rodriguez-Estrada[3], Marina Pérez-Llorca[4], Sergi Munné-Bosch[4], Krzysztof Mikołajczak[1] *

**1** Institute of Plant Genetics, Polish Academy of Sciences, Poznań, Poland, **2** University of Turin, Grugliasco, Italy, **3** University of Bologna, Bologna, Italy, **4** University of Barcelona, Barcelona, Spain

\* kmik@igr.poznan.pl

**Data Availability Statement:** All data generated and analysed during this study are included in the published article and its supplementary information

## Abstract

The advanced molecular tools provide critical inputs in uncovering the regulatory mechanisms underlying plants' adaptation to abiotic stress. Presented holistic studies were done on the barley crown tissue being essential for plant performance under various environmental stimuli. To investigate the effect of brassinosteroids (BRs), the known players in stress management, on molecular response of this tissue to drought, the genotypes with different BRs signal transduction efficiency were employed. Large-scale transcriptomic and proteomic profiling confirmed the specific re-modeling of behavior of the BRs-insensitive barley *uzu1.a* mutant under drought. On the other hand, a set of genes expressed independently of the genotype was identified, including dehydrin encoding genes. This study also uncovered the candidate genes to be linkers of phytohormones crosstalk. Importantly, we detected the converging upregulation of several proteins and encoding genes under drought, including late embryogenesis abundant proteins and chaperones; they represent a promising target for cereals' improvement. Moreover, the greatest variation between genotypes in accumulation of BRs in the crown tissue exposed to drought was observed for castasterone. Presented multi-omics, high-throughput results enhanced the understanding of molecular response to drought in crown tissue. The new insight was provided into the relationships between gene expression, protein and phytohormone content in barley plants of different BRs signaling.

## Introduction

Scientists are observing changes in the Earth's climate in every region and across the whole climate system. One of the climatic extremes is drought, which is a serious threat to the environment. Unfortunately, climate change exacerbates droughts by making them more frequent, longer, and more severe. Over the last few decades, omic high-throughput techniques have

files. Additionally, RNA-seq data used in this paper are available in the ArrayExpress repository, accession number E-MTAB-12773 (https://www.ebi.ac.uk/biostudies/arrayexpress/studies/E-MTAB-12773). Public, open access database EnsemblPlants (https://plants.ensembl.org) was also used for raw data processing.

**Funding:** This work was supported by the National Science Centre, Poland, project Opus 12 no. 2016/23/B/NZ9/03548. The funders had no role in study design, data collection and analysis, decision to publish, or preparation of the manuscript.

**Competing interests:** The authors have declared that no competing interests exist.

been employed to simultaneously identify and quantify changes in gene expression (transcriptomics), proteins (proteomics), metabolites (metabolomics) or elemental composition (ionomics) to decipher adaptive changes in response to biotic or abiotic stresses [1–3]. Next generation sequencing (NGS) technologies providing high-throughput analyses at the genome level have been recently developed. This has accelerated the recognition of many genes controlling the plant behavior, including barley, and along with available reference genome being updated recently, barley has become a convenient organism to examine and understand many different genetic mechanisms including the response to abiotic stresses. Since the genomes of barley and other *Triticeae* species are colinear, this allows to extend the knowledge of barley genome to other genus of plants [4–7].

Although the importance of the crown region (first node above the seed) is known as the key organ for cereal survival, the crown's transcriptomic, proteomic and metabolomic responses to abiotic stresses have not been extensively studied in barley, so far. Such studies seem particularly important due to the properties of meristems located in crowns, affecting regeneration of roots and shoots after exposure to different stresses. Meanwhile, few studies have been conducted in this tissue [8,9]. In one of the newest, Mikołajczak et al. [10] investigated plant modifications and crown-specific transcriptome re-modeling evoked by elevated temperature in two barley *sdw1* semi-dwarf mutants, characterized by partial and total loss of function of the *HvGA20ox2* gene. Authors demonstrated that disturbances in gibberellins (GAs) metabolism effected on heat-induced response of barley and the reaction was dependent on the *sdw1* allele variation. Moreover, they suggested that the disfunction of *HvGA20ox2* may be partially compensated by expression of its paralogue, i.e., the *HvGA20ox4* gene.

Phytohormones are an important class of compounds which regulate various aspects of plant development and responses to adverse environmental conditions. Besides the established classical groups of phytohormones, brassinosteroids (BRs) have emerged as prominent ones in recent times [11]. It has been reported that some of the components of BRs signaling pathway act as multifunctional factors involved in other signaling networks controlling diverse physiological processes. Regulation of some of these processes is mediated through a crosstalk between BRs signalosome and the signaling cascades of other phytohormones [12,13]. These aspects of BRs metabolism have been extensively studied in *A. thaliana* [14–18], but they are much less investigated in monocots, including barley. Despite the fundamental role of BRs in plant functioning, only few studies have examined the effects of BRs on global gene expression regulation in plants [13,19–21]. In barley, the crucial role of BRs in regulating plant growth has been confirmed by molecular characterization of the spontaneous mutant *uzu*, and *HvBRI1* (*Brassinosteroid-Insensitive1*) was identified as a candidate for the *uzu* locus. The *uzu* gene has been introduced into all hull-less barley cultivars in East Asia as an effective dwarf gene for practical use [22]. Semidwarf barley mutants with defects in BRs signaling or biosynthesis are a valuable tool to study the role of these hormones in response to different abiotic stresses, including drought, and they have been suggested as alternative resources for barley improvement in breeding programs [23]. However, there are inconsistent reports about BRs positive/negative role in regulating plant development in various environmental conditions. Lv and Li [15] reported that BRs have a negative impact on floral transition by preventing *Arabidopsis* reproductive development, which begins with the floral transition and ends with the main stem and flowers. Another study reports that BRs can cause inhibition of root growth at higher concentrations in maize [24]. On the other hand, it has been demonstrated that exogenous application of BRs improved the tolerance of various plant species to multiple stresses [25]. Also, Sahni et al. [26] proved that enhanced expression of BRs biosynthetic genes resulted in increased drought tolerance in *Brassica napus*. Among all BRs known to date, castasterone and brassinolide are the most significant BRs because of their higher biological activity and

widespread distribution in plants [27]. Like BRs, sterols, in addition to their role as structural components of cell membranes, are essential for plant growth and development. Consistently with their role as BRs precursors, sterols possess many functions in common with BRs, and *Arabidopsis* mutants impaired in phytosterol biosynthesis generally display BR-deficient-like phenotypes, including dwarfism and reduced fertility [28]. It appears that since BRs promote cell division, a higher level of BRs in the crown should facilitate tillering. It has been established that the overexpression of *miR397* activates the BRs response and leads to an increase in tillering in rice [29]. It was also found that BRs promote bud outgrowth in tomato, and BRs signaling integrates multiple pathways that control shoot branching [30]. Noteworthy, the growth-promoting phytohormones BRs have distinct roles in regulating tillering in rice and rosette branching in *Arabidopsis*, respectively. In *Arabidopsis*, BRs do not regulate the primary branch number [31]; however in rice, BRs significantly promote tillering [32]. Given that the relationship between BRs and tillering is understood, it should be noted that the functioning of BRs in the crown of barley has not been previously investigated, so multiomic studies were designed in the crown of genotypes differentiated by their efficiency in BRs signaling mediated by the BRI1 receptor.

The aim of the present study was the elucidation of drought-induced molecular events occurred in crown tissue of barley. Secondly, we were interested to uncover how disturbance of brassinosteroids signaling may affect the barley multivariate response to drought. Herein, the *uzu1.a* barley mutant with perturbed perception of brassinosteroids mediated by BRI1 receptor was investigated. Experiments were designed from a multi-perspective approach ranging from transcriptomics and proteomics, through phytohormones level determination and phenotypic properties to enrich the knowledge about better adaptations of plants to stressful growth conditions.

## Materials and methods

### Plant material and phenotyping characteristic

Plant material consisted of cv. Bowman (BW, wild type) and its near isogenic line (NIL) BW885 (BRs-insensitive). Bowman is a two-rowed spring barley (*Hordeum vulgare* L.) cultivar and was used for development of numerous NILs by back-crossings [33]. BW885 (*uzu1.a*) carries a missense mutation in the brassinosteroid-signaling gene *HvBRI1* (brassinosteroid-insensitive 1) encoding hormone receptor and exhibits a semi-dwarf phenotype as well as the lodging resistance. It has been demonstrated by Dockter et al. [34] that intorgressed region containing the *uzu1.a* locus in 3H chromosome of BW885 had 6.49 cM. Seeds of the barley accessions (Bowman–NGB20079, BW885– NGB20787) were obtained from the Nordic Genetic Resource Center (NordGen, Alnarp, Sweden).

Mature plants were harvested manually and plant structure and yield components were evaluated with the distinction between main and lateral stems; major phenological stages were also noted. In total, 26 traits (T1-T26) were scored as indicated in the S8B Table. All these studies were based on three biological replicates, each with five plants per one pot and presented by average trait values.

### Abiotic stress application

Drought stress experiment was conducted in the growth chambers under fully controlled conditions according to Kuczyńska et al. [35] with modifications. Seeds were sown in pots (H-LSR 4.5 L; 21 cm in diameter and 20 cm in height) filled with a mixture of loamy soil and peat (3:1, w/w) and plants were cultivated in optimal conditions: soil moisture above 70% field water capacity (FWC), 22˚C/16˚C day/night, air humidity of 60–70%, a photoperiod of 16/8 h of

light/dark. Drought (D) was imposed at the tillering stage (23–26 of BBCH code) and maintained for 10 days. Water scarcity was established at 20% FWC [36] and soil moisture in each pot was controlled gravimetrically by weighing and volumetrically (if necessary) using the FOM/mts device [37,38]. In parallel, the control irrigation experiment (C) was carried out. The number of pots was established so as to contribute with an adequate amount of plant material for all molecular studies along with phenotyping.

Biological samples of barley crown tissue for molecular analyses were collected at early stage of drought (3rd day of stress, 3D) and at the end of stress application–late drought (10th day of stress, 10D). Each replication consisted of crown samples collected from three plants per one pot.

## Single nucleotide polymorphism (SNP) genotyping

Genomic DNA was extracted from 2-week-old leaves using the Wizard® Genomic DNA Isolation Kit (Promega, Madison, WI, USA), according to the manufacturer's instructions. DNA quality and concentration were evaluated using a NanoDrop2000 spectrophotometer (Thermo Fisher Scientific, Waltham, CA, USA) at ratios of >1.8 for 260/280 and 260/230. DNA samples were diluted to 50 ng/μL using molecular biology-grade water (Merck, Darmstadt, Germany). Frozen DNA solutions (20 μL) were delivered for genotyping.

The overall genetic composition of the barley genotypes was investigated using two approaches: single nucleotide polymorphism (SNP) calling from RNA-seq data (described below) and genotyping using a 50k Illumina Infinium iSelect SNP array [39] conducted by Trait-Genetics GmbH (Gatersleben, Germany).

## Whole-genome expression analysis

Barley crown tissue was sampled from both D and C variants at two time points (3D and 10D), as mentioned above. Next, they were frozen using liquid nitrogen and stored at -80˚C until analysis. RNA from crown tissue was extracted from four replicates according to Mikołajczak et al. [40]; the methodology included the following steps: total RNA extraction (Qiagen RNeasy Plant Mini Kit, Hilden, Germany), removal of genomic DNA contamination during (on-column DNase digestion, RNase-Free DNase Set, Qiagen) and after RNA isolation (DNase Max Kit, Qiagen). RNA quantity, quality and integrity were determined based on the study by Mikołajczak et al. [10]. Construction of cDNA library (TruSeq stranded mRNA) and mRNA sequencing were commissioned to Macrogen Inc. (Seoul, Republic of Korea) that employed an Illumina NovaSeq6000 platform with a 150 bp paired-end configuration and the numbers of read pairs from 23.2 to 34.4 M per sample.

## Proteomic profiling

Barley crown tissue (about 100 mg) gathered from D and C conditions variants at two time points (3D and 10D) was frozen with liquid nitrogen, ground to a fine powder and stored at -80˚C. Protein extraction was performed according to Hurkman-Tanaka protocol [41] in three replicates. Pierce™ bicinchoninic acid (BCA) protein Assay Kit (ThermoFisher Scientific) was employed for protein quantification. Furthermore, peptide solution was pre-treated with 100 mM dithiothreitol (DTT) for 5 min at 95˚C and 100 mM iodoacetamide for 20 min at room temperature; afterwards, it was subjected to 'in-solution' digestion with trypsin solution (Sequencing Grade Modified Trypsin, Promega) overnight. Proteomic profiling was conducted employing a Dionex UltiMate 3000 RSLC liquid chromatograph coupled with Q Exactive high-resolution mass spectrometer with an Orbitrap mass analyzer equipped with H-ESI ion source (ThermoFisher Scientific). Two mobile phases were used: A–water

containing 0.1% formic acid (v/v) (LC-MS grade, Merck) and B–acetonitrile (LC-MS grade, Merck). The following linear gradient was employed: 0 min, 5% B; 5 min, 5% B; 160 min,70% B; 160 min, 95% B; 170 min,95%; 170 min, 5% B; 180 min, 5% B. Next, 5 μL of sample were injected onto Acclaim PepMap RSL C18 column (75 μm × 250 mm, 3 μm, ThermoFisher Scientific) for the chromatographic separation with 0.3 μL/min flow rate. The operating mode of the mass spectrometer included the following parameters: positive-ion mode (spray voltage 1.5 kV, capillary temperature 250˚C, S-lens radio frequency (RF) level 50.0); full scan with a 350–2000 $m/z$ range (– 70,000, automatic gain control (AGC) target– 1e6, maximum injection time (IT)– 100 ms); settings of data dependent mode (resolution 17.500, AGC target– 5e4, maximum IT– 100 ms, isolation window– 2.0 $m/z$), and normalized collision energy. Data were acquired and processed by using Xcalibur 4.0 software (ThermoFisher Scientific).

## Determination of phytohormone and sterol composition

Plant material (about 100 mg) of barley crown tissue was collected from drought and optimal conditions variants at two time points (3D, 10D) along with one more time point obtained one day before stress application (0D). Collected material was frozen with liquid nitrogen, ground to a fine powder using a ball mill (Retsch MM 400) and stored at -80˚C until analyses, which were performed in three biological replications. Brassinosteroids, including brassinolide (BL), castasterone (CS), and cathasterone (CT) were quantified using high performance liquid chromatography coupled to electrospray ionization tandem spectrometry (HPLC/ESI-MS/MS) as described in Setsungern et al. [42]. Extracts were obtained by vortexing, ultrasonication for 30 min, and centrifugation (9500 $g$, 15 min, 4˚C) of the mixture of plant material powder and extraction solvent (methanol and 1% glacial acetic acid) containing labelled standards ($d_3$ –BL, $d_3$ –CS, $d_3$ – CT). The extracts were then filtered with 0.22 μm PTFE filters (Phenomenex, Torrance, CA, USA) prior to injection into the HPLC/ESI-MS/MS system. The HPLC system consisted of an Agilent 1200 HPLC™ System (California, United States) binary pump equipped with an autosampler and the mass spectrometer consisted of a 4000 Q TRAP triple quadrupole mass spectrometer (AB Sciex, Washington, United States). A HALO™ C18 (Advanced Materials Technology, Wilmington, United States) column (2.1 × 75 mm, 2.7 μm) was used for the analysis of the extracts and the injection volume 5 μl. Gradient elution was carried out with water containing 0.05% acetic acid as solvent A and acetonitrile with 0.05% acetic acid as solvent B, maintaining a constant flow rate of 350 μl/min. The gradient profile (time [min], % A) was set as follows: (0, 70), (2, 20), (8, 5), (9, 70), and (14, 70). Measurements were conducted by multiple reaction monitoring (MRM) in positive ion mode. Quantification was based on standard curves generated from commercially available brassinosteroid standards and their corresponding deuterated forms, all sourced from Olchemim (Olomouc, Czech Republic). Optimized MRM parameters are listed in S1 Table, while extracted ion chromatograms (XIC) of MRM and enhanced product ion (EPI) spectra of the brassinosteroids analyzed in barley samples are shown in S1 Fig.

Total sterols were isolated from barley crown tissue and then analyzed by Fast-GC/MS (GC Shimadzu QP 2010 Plus; Kyoto, Japan) according to Kuczyńska et al. [35]. TMS-sterols and stanols were recognized by comparing the mass spectra with those of corresponding purchased (Sigma) and synthesized chemical standards, respectively [43]; they were quantified by using their corresponding characteristic ions ($m/z$): 343 (campesterol), 83 (stigmasterol), 396 (sitosterol), 488 (sitostanol), and 459 (campestanol).

## Data analysis

The IBSC_v2 Hordeum vulgare (Ensembl Plants rel. 48) genome assembly was used as a reference for SNP and gene expression analyses. mRNA-seq data analysis was performed as in our

previous study [40]. Briefly, after adapter removal and quality trimming, TopHat 2.1.1 was used for mRNA-seq reads mapping [44] (parameters: maximum no. of mismatches = 1,—no-mixed,—no-discordant; the mapping rate was 80.3–85.7%), whereas DESeq2 1.22.2 [45] was employed for differential expression analysis (with conditions: mean number of mapped reads > 5; |log2(FC)| > 2; FDR < 0.01). SNP calling was performed using the samtools/bcftools pipeline [46] (filtering parameters: %QUAL > 40, MAF > 0.10, DP > 80). GOfuncR 1.18.0 package was used for Gene Ontology terms overrepresentation analysis (with FDR < 0.01). 'Venn' 1.11 package in R was used to create Venn diagrams, whereas Ensembl Variant Effect Predictor (VEP) tool was used for the SNP protein translation effects prediction [47,48]. Genstat 19 was employed for statistical analyses not attributed to other tools [49].

Data analysis for proteomic profiling were performed using MaxQuant 2 (1.5.3.1) and Perseus (1.4.1.3) software together with the commercial software Proteome Discoverer 2.2 (ThermoFisher Scientific). Protein libraries were explored handling the SequestHT tool for proteins prepared for *Hordeum vulgare* in the UniProt database. Differential expression of proteins was declared at FDR < 0.05.

Phytohormone and sterol data were subjected to analysis of variance (ANOVA) in the model containing fixed effects of genotype, treatment and time point, and of their pair-wise interactions. Phenotypic data were analyzed by ANOVA with effects of genotype, treatment, and of their interaction.

## Results

### Single nucleotide polymorphism (SNP)

Two methods of SNP finding were employed to compare the genetic composition of BW and BW885. Out of 3.130 polymorphic markers between genotypes, 1.863 were homozygous in both forms (S2 Table). 160 SNPs were detected by both applied methods at the same position; in all those cases the readings were consistent. The greatest accumulation of homozygous SNPs was observed in four clear hotspots in chromosomes: 3H (region A, 491084467–536962525, containing 209 polymorphic genes), 1H (region B, 13575292–25339880, 60 genes), 2H (region C, 3096990–10215806, 52 genes), and 6H (region D, 554034555–561641532, 45 genes) (Fig 1). Overall, 21 homozygous SNPs of 'HIGH' protein translation effect were found by VEP tool (Ensembl Plants), especially in stop codons ('stop gained' or 'stop lost' variants'). There were 22 homozygous SNPs (50k Chip) located in 'upstream gene variant' of 17 genes with 'MODIFIER' effect (S2 Table).

### Differential gene expression

Overall, 2.481 differentially expressed genes (DEGs) were identified in at least one of the eight defined comparisons (S3 Table), including 136 polymorphic genes between genotypes. Genotypic DEGs (i.e., revealed between genotypes, BW885 vs. BW) at 3D were more numerous in control conditions than in drought, but the opposite situation was observed for 10D, with the greatest number of shared DEGs between C and D at 3D (Table 1A, Fig 2A). The number of genes downregulated in BW885 was larger than the number of upregulated ones in all variants, except for D at 10D. Considering time effect, more genotypic DEGs shared between 3D and 10D were found under drought than in control conditions (Fig 2A). In general, greater number of specific genotypic DEGs was identified in C at 3D and in D at 10D than in other experimental variants.

Substantially more gene expression changes were induced by drought (D vs. C comparison) at 10D than at 3D, with a maximum number of modifications in BW at 10D (Table 1B); also, the ranges of differential expression effects were wider at 10D than at 3D. Again, the number

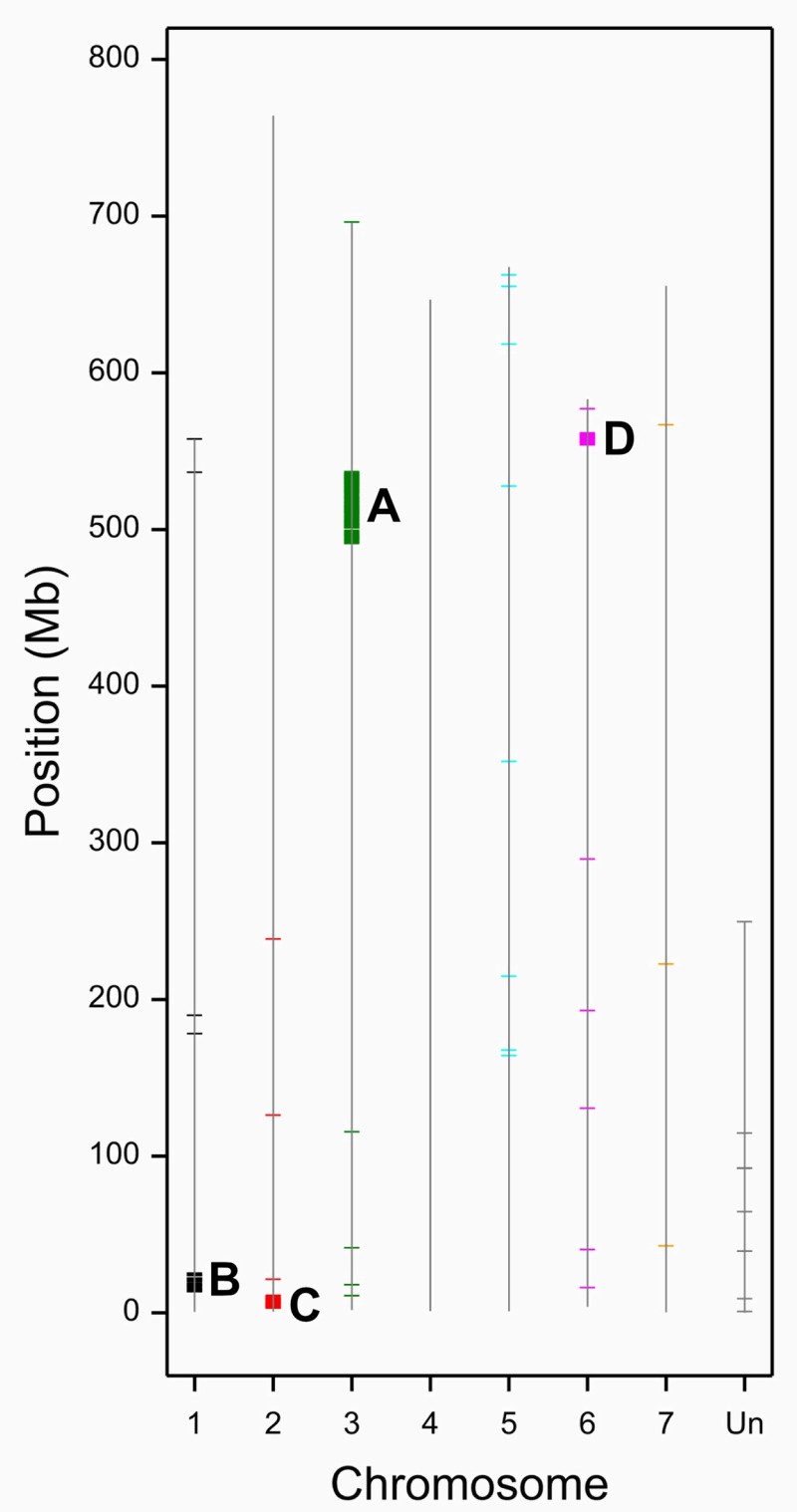

**Fig 1. Localization of SNPs observed between BW and BW885 in barley chromosomes.** Rectangles mark hotspots (A-D) with more than 4 SNPs per 1 Mb.

**Table 1. Comparison of the number of differentially expressed genes (DEGs) between genotypes (A) and between treatments (B).**

| A–Genotypic DEGs (BW885 vs. BW) | | | | |
|---|---|---|---|---|
| Regulation status | 3D | | 10D | |
| | C | D | C | D |
| Down (-1) | 274 | 169 | 114 | 207 |
| Up (1) | 164 | 60 | 17 | 276 |
| Total | 438 | 229 | 131 | 483 |
| log2FC min, max | -9.3, 6.1 | -9.5, 5.3 | -9.6, 5.4 | -9.4, 5.6 |

| B–DEGs in comparison of treatments D vs. C | | | | |
|---|---|---|---|---|
| Regulation status | Bowman | | BW885 | |
| | 3D | 10D | 3D | 10D |
| Down (-1) | 179 | 1093 | 136 | 113 |
| Up (1) | 11 | 702 | 89 | 592 |
| Total | 190 | 1795 | 225 | 705 |
| log2FC min, max | -4.5, 2.9 | -7.1, 10.6 | -3.9, 4.6 | -5.5, 8.8 |

of genes downregulated under drought was larger than the upregulated ones in all variants, except for BW885 at 10D. Approx. 7-fold more drought-induced DEGs were shared between genotypes at 10D compared to 3D, which represented ca. 63% and 25% of all DEGs in BW885 and BW at 10D, respectively (Fig 2B). Among genotype-specific DEGs under drought treatment (D vs. C, 10D), we found six genes annotated to ATP-binding cassette (ABC) transporters; those identified in BW885 were upregulated, whereas BW-specific genes had reduced expression in response to stress, with one exception (S3 Table). Interestingly, the genotypic comparison also revealed the difference between genotypes in regulation of ABC transporters-

**Fig 2.** Venn diagrams visualizing the number of differentially expressed genes in crown tissue in: (A) BW885 vs. BW comparison, from left to right DEGs specific and shared between control (C) and drought (D) at 3D and 10D, and between time points in C and in D conditions; (B) D vs. C treatments comparison, from left to right DEGs specific and shared between genotypes at 3D and 10D, and between time points in Bowman and in BW885.

related genes at 10D; in particular, three and four DEGs had decreased and increased expression in control and drought conditions, respectively, in BW885 compared to BW. Considering time effect, a similar number of DEGs was shared between 3D and 10D for both genotypes, despite 2-fold more DEGs detected for BW than for BW885 at 3D vs. 10D comparison (Fig 2B).

Additionally, we classified DEGs according to pairwise contrasts within: (i) genotype comparison to investigate the interaction of genotypic DEGs effect with treatment (i.e. water regimes D and C) and time (3D and 10D) (Table 2A), and (ii) treatment comparison to investigate the interaction of DEGs effect (D vs. C) with genotype (BW885 and BW) and time (3D and 10D) (Table 2B). On this basis, DEGs showing opposite regulation status between pairwise contrasts were indicated (Fig 3). Four genes (HORVU4Hr1G076000, HORVU5Hr1G118200, HORVU7Hr1G119930, HORVU7Hr1G120060) in BW885 were identified whose direction of expression was inverted over the time of plant exposition to drought: downregulation at early drought (3D) but upregulation at late drought (10D) in relation to control watering. Three genes (HORVU3Hr1G074950, HORVU3Hr1G105420, HORVU7Hr1G051310) with contrasted expression at late drought (10D) were found between genotypes: they were downregulated in BW and upregulated in mutant BW885. Lastly, two DEGs (HORVU2Hr1G117610, HORVU3Hr1G074950) between genotypes were found, whose expression status was changed from negative in control to positive at late drought (10D).

Identified DEGs were functionally interpreted using Gene Ontology (GO) enrichment analysis (S4 Table). About 2.5-fold more significant cases of enriched GO terms came out for DEGs identified in D vs. C than in genotype comparison. DEGs affected by drought revealed higher number of enriched GO terms at 10D than at 3D for both genotypes. Importantly, all GO terms (biological process) related to regulation or negative regulation of different processes (in total 16 terms, e.g. regulation of proteolysis, negative regulation of nitrogen compound metabolic process) were enriched specifically in BW885 at 10D and the set of genes

**Table 2. DEGs classified according to pairwise contrasts within genotype (A) and treatment (B) comparison.**

**BW885 vs. BW**

| | | Number of DEGs in 3D, D | | | Number of DEGs in 10D, C | | |
|---|---|---|---|---|---|---|---|
| | | down | nc | up | down | nc | up |
| Number of DEGs in 3D, C | down | 131 | 143 | 0 | 42 | 232 | 0 |
| | nc | 38 | 33522 | 36 | 72 | 33517 | 7 |
| | up | 0 | 140 | 24 | 0 | 154 | 10 |
| Number of DEGs in 10D, D | down | 108 | 61 | 0 | 40 | 72 | 2 |
| | nc | 99 | 33454 | 252 | 167 | 33473 | 263 |
| | up | 0 | 36 | 24 | 0 | 6 | 11 |

**>D vs. C**

| | | Number of DEGs in BW, 10D | | | Number of DEGs in BW885, 3D | | |
|---|---|---|---|---|---|---|---|
| | | down | nc | up | down | nc | up |
| Number of DEGs in BW, 3D | down | 99 | 80 | 0 | 64 | 115 | 0 |
| | nc | 994 | 32156 | 694 | 72 | 33685 | 87 |
| | up | 0 | 3 | 8 | 0 | 9 | 2 |
| Number of DEGs in BW885, 10D | down | 89 | 1001 | 3 | 11 | 121 | 4 |
| | nc | 24 | 31978 | 237 | 102 | 33197 | 510 |
| | up | 113 | 33329 | 592 | 0 | 11 | 78 |

nc–no significant change of expression.

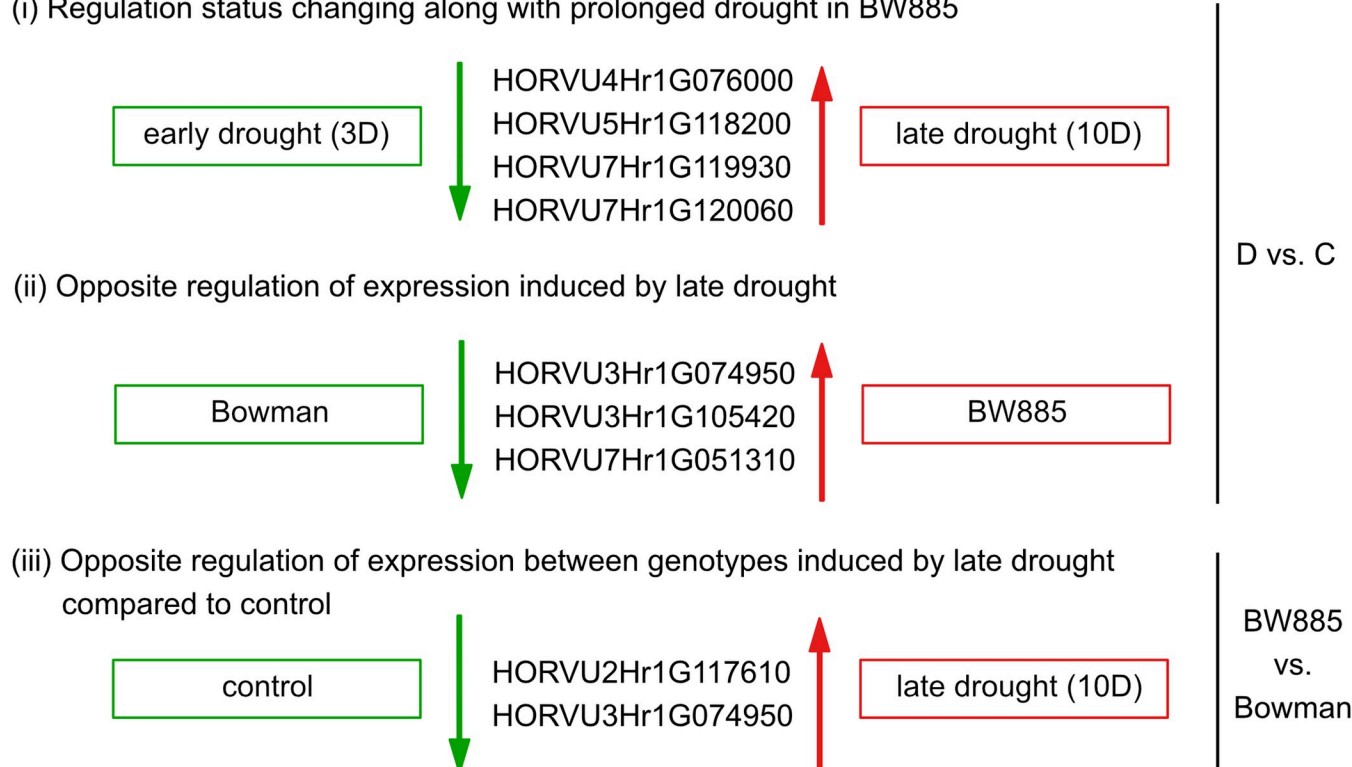

**Fig 3. Differentially expressed genes showing opposite regulation status between pairwise contrasts (↓ - downregulation; ↑ - upregulation).**

assigned to those terms was similar (15–18 genes). All of them were overexpressed under drought compared to control at 10D. Similarly, all DEGs assigned to terms 'lipid transport' and 'lipid localization', being exclusively enriched in BW885 at 10D, were upregulated. There were also several BW-specific enriched terms, including 'response to oxidative stress', where all assigned DEGs were downregulated under drought with four exceptions (in total 41 genes). Unexpectedly, terms 'response to water deprivation' and 'response to abiotic stimulus' were not enriched in any case within D vs. C comparison. In turn, GO term 'response to stress' was overrepresented at 10D under drought compared to control in both genotypes, with about 2 times more genes assigned to the term in BW than in BW885. Interestingly, approx. 2/3 of DEGs in BW had reduced expression in response to drought, whereas in BW885 almost 90% of DEGs were upregulated. Few terms associated with oxidoreductase activity (molecular function) were mainly enriched in BW in both time points, but definitely more DEGs were assigned to these GO terms at 10D than at 3D; in general, they were downregulated, especially at 3D. In the fraction of DEGs between genotypes (BW885 vs. BW), most of enriched GO terms were found in C at 3D. Noteworthy, all terms (23) within cellular component category were enriched only in C at 3D, with two exceptions.

## Expression analysis of phytohormone-related genes

To learn more about behavior of genes associated with main phytohormone classes affecting branching, namely brassinosteroids, gibberellins, and strigolactones (SLs), under drought, GO terms related to their metabolism and signaling were identified (S5 Table). Next, 64 genes corresponding to GO terms linked to brassinosteroid were found in our transcript set and seven

of them were DEGs (one contained SNP) in at least one comparison (S6 Table). The biggest number of DEGs (four genes) was found between drought and control in BW at 10D and all of them were downregulated, and belonged to cytochrome P450 family, except for HOR-VU2Hr1G115240 encoding bHLH transcription factor. Gene HORVU4Hr1G032830 showed increased expression in BW885 compared to BW under drought, regardless of time points, which encoded again cytochrome P450. No significant expression changes in the *HvBRI1* (*uzu1.a*, HORVU3Hr1G068000) gene in crown tissue of BW and BW885 was observed in our experiment. By analogy, GAs- and SLs-related DEGs were identified in transcripts set. Out of 53 and 6 genes associated with gibberellin and strigolactone, 8 and 1 were found to be DEGs, respectively (S6 Table). Two of GAs-related genes had the same regulation status in both genotypes in D vs. C comparison; however, HORVU3Hr1G022840 (*HvGA3ox2*) and HOR-VU5Hr1G124120 (*HvGA20ox1*) were down- and upregulated at 10D, respectively. Drought treatment resulted in decreased expression of HORVU1Hr1G069310 and HOR-VU2Hr1G004620 in BW at 10D, but parallelly HORVU4Hr1G058070 was overexpressed in BW at 3D. On the other hand, HORVU1Hr1G086810 (encoding GA2-oxidase) was upregulated only in BW885 in response to D at 10D, even though the gene expression was reduced in mutant in relation to BW in control conditions (at 3D). Gene HORVU6Hr1G028790 (encoding TFs of WRKY family) was not affected by drought in both genotypes, but its overexpression was observed in BW885 compared to BW in D at 10D. Noteworthy, gene HORVU3Hr1G019840, whose expression was increased by drought in BW at 10D, was annotated to various hormones including, apart from gibberellin, also response to auxin, ethylene and jasmonic acid. Single DEGs (*HvHTD3*) associated with SLs was negatively induced by late drought (10D) in both genotypes.

In order to expand the panel of DEGs associated with phytohormones, we used KEGG and Gramene databases of functional pathways for identification of genes corresponding to hormones of interest. This approach revealed 6, 16 and 4 additional DEGs associated with BRs (in total 10 DEGs), GAs (in total 17 DEGs) and SLs (in total 5 DEGs), respectively (S6 Table), which were not found through GO annotation. Drought resulted in reduced expression of BRs-related genes, especially for BW at 10D; however, in HORVU2Hr1G104040, the upregulation occurred. Interestingly, this gene was involved in both BRs and GAs signaling pathways. In contrast HORVU6Hr1G068980, another gene associated with BRs and GAs signaling, reduced expression in BW under drought at 10D. Also, we found that the gene HOR-VU5Hr1G046390 was involved in signaling of brassinosteroid and jasmonic acid, whose expression was specifically decreased in BW885 in D at 3D. Most of GAs-related DEGs (14) were involved in gibberellin signaling pathway, again, mainly under drought in BW at 10D. Noteworthy, there was similar number of both down- and upregulated genes associated with GAs signaling, and one, HORVU5Hr1G103770 (upregulation), was specific for BW885 at 10D. Apart from downregulation of HORVU3Hr1G022840 (GAs biosynthesis) in both genotypes under drought at 10D, as previously mentioned, upregulation of GAs signaling gene (HORVU3Hr1G024660) was revealed at the same contrasts. There was one genotypic DEG (HORVU2Hr1G104160) involved in GAs and JA signaling, whose expression was enhanced in BW885 compared to BW in D at 10D. Drought reduced the expression of all SLs-related DEGs in BW at 10D except for HORVU4Hr1G079620 (SLs biosynthesis), whose expression was not modified in BW but enhanced in BW885 at 10D. DEGs associated with SLs biosynthesis, HORVU3Hr1G071170 (identified also by GO annotation) and HORVU6Hr1G040170, reduced expression in response to drought also in BW885 at 10D, whereas all DEGs involved in SLs signaling were significant only for BW at 10D under drought.

## Expression analysis of genes related to drought and oxidative stress

Finally, to better understand the reaction of genes strictly related to drought, 34 DEGs (six contained SNPs) annotated to response to drought/water deprivation (S6 Table) were revealed using GO terms listed in S5 Table. Twenty of them were affected by drought and all DEGs (9) annotated to dehydrin were overexpressed under drought at 10D in both genotypes; in parallel, most of dehydrin encoding genes were also upregulated in D at 3D in BW885. Two DEGs belonging to annexin superfamily were found to have genotype-specific reaction under drought, i.e. HORVU1Hr1G016080 was downregulated in BW in D at 10D, whereas HOR-VU4Hr1G074130 was upregulated in BW885 in D at 10D. Moreover, DEGs associated with transcription factors were identified in this subset of transcripts: HORVU3Hr1G084360 encoding basic leucine zipper (bZIP) transcription factor (*HvABF1*) showed BW-specific over-expression under drought at 10D, HORVU4Hr1G052490 encoding SANT/Myb domain reduced expression in BW under drought at 10D, and HORVU6Hr1G028790 (WRKY domain) was described above to be related to gibberellin. We found 12 genotypic DEG assigned to histone H4 and they were all upregulated in BW885 compared to BW in C at 3D (S6 Table).

Since drought is usually accompanied by oxidative stress, also regulation status of genes corresponding to this stressor (47 DEGs including one polymorphic gene) was analyzed according to the above-mentioned approach, using corresponding GO terms listed in S5 Table. All DEGs were described to encode peroxidase and were distributed onto seven chromosomes, the most numerous being in 2H and 7H (S6 Table). In general, they were affected by drought at 10D, and almost 7-fold more DEGs were downregulated in BW than in BW885. On the other hand, BW885 had more upregulated genes (4) than BW (1) under drought compared to control at 10D; gene HORVU7Hr1G083550 was overexpressed in both genotypes (D, 10D). One gene related to oxidative stress (HORVU3Hr1G074950) exhibited opposite regulation status between genotypes in response to drought at 10D and it was also a genotypic DEG, as indicated previously (Fig 3). Overall, we found 13 genotypic DEGs related to peroxidase whose expression was greater in BW885 than in BW in D at 10D and none at 3D.

## Differentially abundant proteins (DAPs) and comparison of protein abundance vs. gene expression

To identify drought-induced alterations in protein accumulation we compared their level in crown tissue exposed to drought compared to control condition (D vs. C) in BW885 and BW. Proteomic profiling revealed 3.645 proteins, of which 565 were differentially accumulated in at least one of the four contrasts within D vs. C comparison (S7 Table). Despite the total number of DAPs being significantly smaller than the number of DEGs detected in D vs. C comparisons in general, their pattern of expression changes was similar. More DAPs were found at 10D than at 3D, especially in BW (above 4-fold). The number of downregulated proteins was larger than the number of upregulated ones in Bowman at 10D and in BW885 at 3D (Table 3).

**Table 3. Number of differentially accumulated proteins (DAPs) in comparison between treatments (D vs. C) for Bowman and BW885 at two time points (3D, 10D).**

| Time point | Bowman | | BW885 | |
|---|---|---|---|---|
| | 3D | 10D | 3D | 10D |
| Down (-1) | 40 | 226 | 58 | 35 |
| Up (1) | 47 | 146 | 43 | 82 |
| Total | 87 | 372 | 101 | 117 |

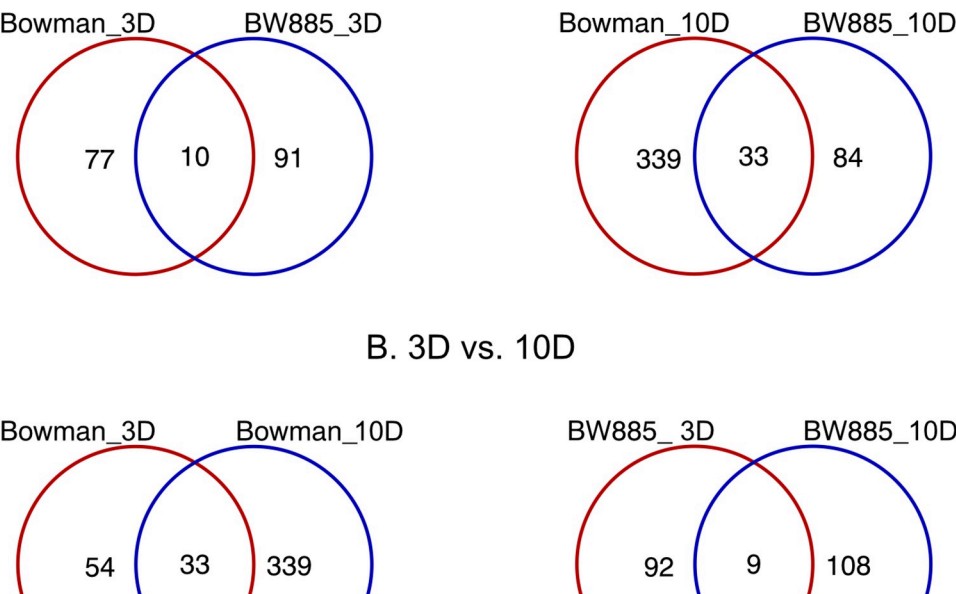

**Fig 4.** Venn diagrams visualizing the number of differentially accumulated proteins in crown tissue in D vs. C comparison, specific and shared between: (A) genotypes at 3D and 10D, (B) time points in Bowman and BW885.

Considering differences between genotypes, 3 times more DAPs were shared between BW885 and BW at 10D than at 3D (Fig 4A). There were 6-fold more specific proteins in BW at 10D compared to 3D. It was similar for BW885, but the difference between the number of proteins specific for each time point was much smaller (Fig 4B). The same tendency was observed for differentially expressed genes. Common DAPs between time points of drought constituted 8% and 4% of all DAPs observed in BW and BW885, respectively.

To investigate the relationship between DAPs and DEGs, gene identifiers (HORVU) were assigned to proteins through the presence of Uniprot TrEMBL protein identifier within annotation of genes. Next, differential expression of genes was compared to differential expression of corresponding proteins. In this way, the most significant relationship between DEGs and corresponding DAPs was found in BW at 10D under drought compared to control conditions (Fig 5). There were 2.305 proteins with associated genes where the expression of both did not significantly change. We found 47 DAPs whose corresponding genes were also differentially expressed in BW at 10D (D vs. C) (Table 4). Out of them, 32 and 11 DAPs were up- and down-regulated together with assigned DEGs, respectively (S7 Table). Within upregulated ones, we observed mostly proteins involved in drought response, especially dehydrins (e.g. dehydrin 7), a few LEA (late-embryogenesis abundant proteins) proteins (including HVA1), chaperones HSP20 (heat shock protein), AAI (alpha amylase inhibitor) domain containing protein and jasmonic acid domain containing protein. Most of DEGs associated with overexpressed DAPs were upregulated also in BW885 at 10D, but with no effects on proteins regulation. However, one of dehydrin (encoded by HORVU3Hr1G089300) and LEA protein (encoded by HOR-VU5Hr1G094680) showed downregulation under drought in BW885 at 10D despite

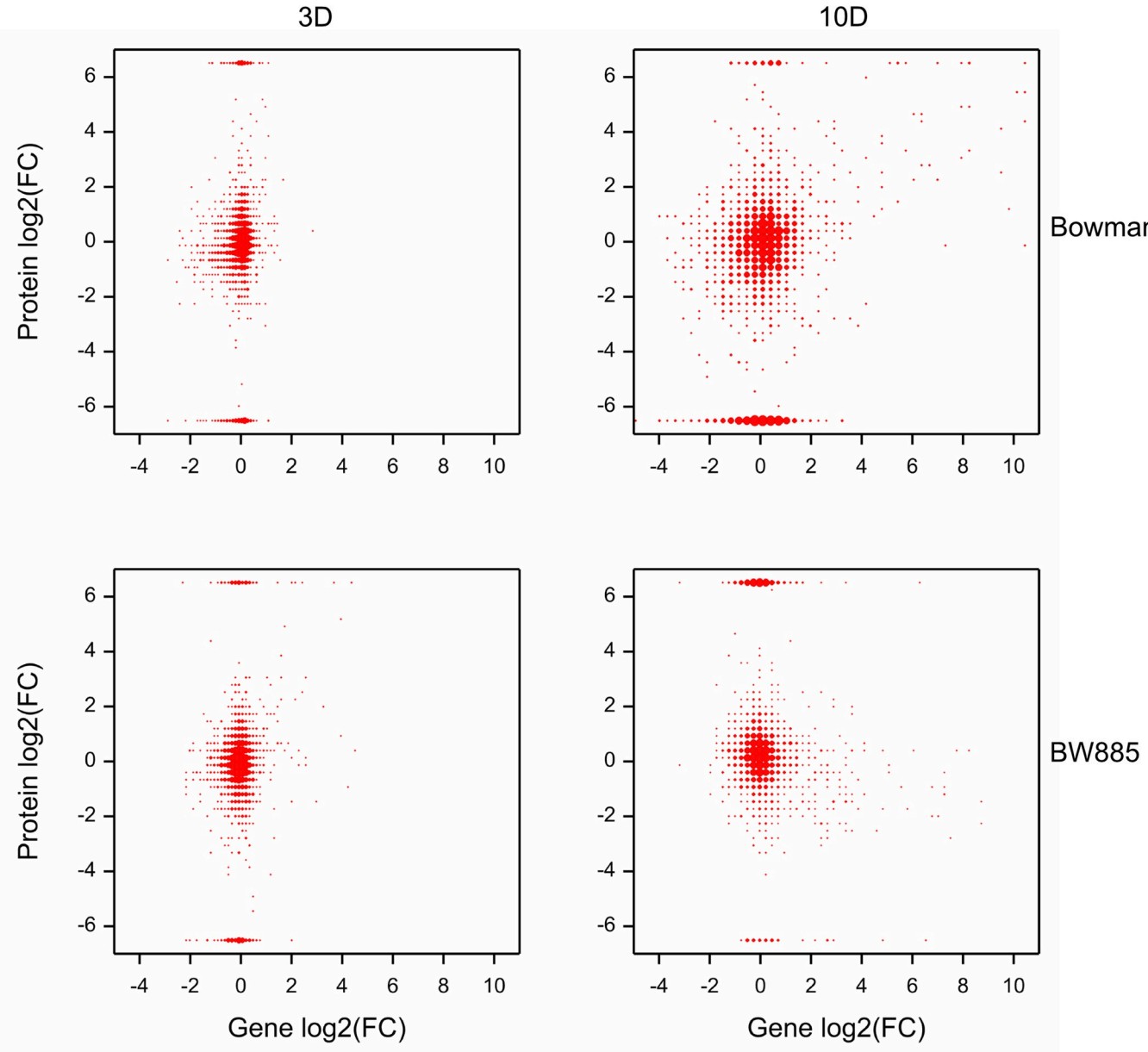

**Fig 5. Density plot of differential expression scores of genes in relation to differential expression scores of proteins in four contrasts of D vs. C comparison.**

**Table 4. Number and regulation status of proteins and corresponding genes in Bowman at 10D in D vs. C comparison; association between regulation of DEG and DAP significant at $P < 0.001$ (chi-square test).**

|  |  | DEG | | Total |
|---|---|---|---|---|
|  |  | No | Yes |  |
| DAP | No | 2305 | 135 | 2440 |
|  | Yes | 229 | 47 | 267 |
| Total | | 2534 | 182 | 2716 |

confirmed overexpression of assigned DEG, and another two dehydrins were upregulated in BW885 at 3D under drought (S7 Table). Among downregulated cases, there were among others: HORVU5Hr1G109040 (annotated to lipid transfer protein) linked to AAI domain containing protein (interestingly, this protein was upregulated in BW885 at 10D under D), HORVU7Hr1G095900 (annotated to cytochrome P450) linked to unnamed protein being involved in tricin biosynthesis according to KEGG/Gramene pathways, HOR-VU3Hr1G090970 (annotated to S-adenosyl-L-methionine-dependent methyltransferase) assigned to uncharacterized protein. There were also two genes (HORVU2Hr1G063510 and HORVU6Hr1G024780) whose encoded protein was associated with NAC5 transcription network (acc. to KEGG/Gramene). In addition, in four cases the regulation status between DAPs and corresponding DEGs was opposite in BW under drought at 10D. No changes were noted in relation to regulation status of proteins whose assigned DEGs were associated with hormones BRs, GAs and SLs.

In the enrichment analysis of the total set of DAPs, as well as those specific to BW at 10D (D vs. C comparison), we used PANTHER Protein Class as annotation data set. None of the terms was significantly (FDR < 0.05) overrepresented.

## Phytohormones and sterols quantification

Three main classes of brassinosteroids were examined three days before stress application, at 3D and at 10D. In control conditions, the highest content of brassinolide in crown tissue was observed at 3D for both genotypes; noteworthy, the level of hormone was about 30% higher in BW than in BW885 (S2 Fig). In turn, castasterone and cathasterone were the most accumulated hormones before drought, and the greatest content was present in BW885 compared to BW. A reverse effect of drought on each BRs class was revealed between genotypes: the content of brassinolide and castasterone increased in BW and decreased in BW885 over drought time, i.e. from the 3rd (3D) to the 10th (10D) day of stress. In parallel, cathasterone level diminished in BW and increased in BW885. Considering the sources of variation, the significant mean effect was found in the following cases: day of stress ($P = 0.012$) for brassinolide; genotype ($P < 0.001$) and genotype × watering interaction ($P = 0.040$) for castasterone; genotype × day of stress ($P = 0.042$) for cathasterone.

Sterols quantification confirmed that sitosterol and campesterol were the major components of the sterol fraction in the crown tissue of both genotypes (S3 Fig). In general, drought resulted in increased accumulation of phytosterols; however, avenasterol decreased in all cases compared to control conditions. Considering the effect of time under drought, the concentration of all sterols decreased at 10D with respect to 3D in both genotypes, except for cholesterol in BW885. From all sources of variation, water treatment mainly affected the phytosterols matrix, being stronger under drought than at optimal watering (S8A Table); the most significant impact was found in sitostanol and avenasterol ($P < 0.001$). Noteworthy, the significant effect of time point (day of stress) and genotype × time point interaction was detected only for avenasterol. A significant genotypic effect was identified only for cholesterol.

## Phenotypes characterization

ANOVA revealed significant effects of genotype and water treatment on most of the observed phenotypic traits (S8B Table). Considering the interaction of genotype × water treatment the significant effect was found for six traits. Overall, the two genotypes differed in analyzed traits in both treatments, i.e., drought and optimal watering (S4A Fig). A much higher increase of tillers number (T1, T2) was detected under drought in Bowman. In contrast, a strong reduction of T11 and T12 was noticed for BW885 in water shortage. Interestingly, reverse effect of

drought was observed for genotypes in relation to T10: reduced for Bowman and increased for BW885. Values (or levels) of traits related to plant height (T3-T6, T8-T11) and length of spike (T13, T17) were greater for Bowman, whereas BW885 was characterized by later heading (T15), and higher values of the rest of spike-related traits (T14-T16, T18), except for T19. Mean effect of drought was found to be negative for most traits, but it was positive for phenological phases (T14-T16) in both phenotypes. In the multivariate sense, the drought effect was bigger for BW885 than for Bowman, and the distance between genotypes was larger in drought than in control conditions (S4B Fig).

## Discussion

Herein, we investigated the multifaceted behavior of barley's crown under exposition to temporal drought in two genotypes varied in efficiency of brassionsteroids perception mediated by the BRI1 receptor.

Our extensive SNP genotyping indicated that BW885, the near isogenic line of cv. Bowman, was genetically more polymorphic in relation to its reference genotype than it has been commonly suggested so far [34]. Thus, to reduce the influence of multiple introgressions (defined as regions containing SNPs), on gene expression analysis the polymorphic DEGs were filtered out from the biological interpretation; only those located in the introgressed region around the *uzu1.a* locus remained. This region was extrapolated roughly based on Dockter et al. [34] study using the *HvBRI1* (*uzu1.a*) gene position (464,679,001–464,682,900; Ensembl Plants), namely, it was the interval 460,879,001–467,372,900 of chromosome 3H (Ensembl Plants); in total, seven DEGs were identified there. Nonetheless, such approach did not substantially affect our interpretation of differential gene expression analysis, since only about 5% of all DEGs contained SNPs. Meanwhile, it should be borne in mind that some removed DEGs can play important regulatory functions. For instance, two DEGs containing SNPs in the "upstream" region of the gene (in promoter sequence putatively) were annotated to heat shock factors (HSFs) regulatory network; in fact, they are suggested to participate in plant growth and stress response [50].

### Differential gene expression

Overall, large-scale transcriptomic study proved the significant discrepancies in expression of drought-inducible genes between Bowman and the *uzu1.a* BRs-signaling mutant. Indeed, abnormalities in BRs perception can affect the activity plethora of genes responsible for plant's functioning and stress signaling, since the broad pleiotropic effect of this hormone is well known [29]. The behavior of genes related to ABC transporters specially drove our interest, because there are very few reports available about barley in this field. It is known that ABC transporters are responsible for the ATP-powered translocation of many substrates across membranes and they are involved in regulation of diverse biological processes, which help plants surviving under adverse environmental conditions [51]. Interestingly, in the newest study by Ying et al. [52], the ABC transporter ABCB19 was proved to participate in BRs export in *Arabidopsis* improving the BRs signaling. Hence, we speculated that different activity of genes encoding ABC transporters between mutant and the wild type may result from the heterogenous BRs signaling in studied genotypes. It can be hypothesized that upregulation of these genes in BW885 compared to BW under drought was the mechanism to increase the BRs transport efficiency to compensate disorders in BRs perception by BRI1 receptor. However, it seems it was ineffective as indicated by the phenotypic observations.

Comparing genotypes during the growth in optimal conditions, we found that approx. 3-fold more genes were differentially expressed at 3D than at 10D, which suggested greater

differences at the transcriptome level of the crown tissue between genotypes in the earlier growth phase, especially with regard to genes assigned to the cellular component category. Meanwhile, the prolonged drought caused larger differentiation of transcriptome between genotypes. For instance, in BW885, the regulatory machinery and transport of lipids were primarily activated (upregulation of corresponding genes), whereas in BW numerous genes associated with the response to oxidative stress were downregulated. Interestingly, although the GO term 'response to stress' was enriched in both genotypes, the assigned genes, again, showed a contrasting behavior between mutant and BW in response to late drought. Taken together, these findings are consistent with the assumption that the extent of damage depends upon crop growth stage and the cell balance under abiotic stresses [53].

Moreover, we found four genes for mutant with opposite regulation status over the time (from down- to upregulation) of plant exposure to drought (i.a. O-methyltransferase family protein, Leucine-rich repeat receptor-like protein kinase family protein, Cathepsin B-like cysteine proteinase 1) with fundamental roles in development, stress response, disease resistance and brassinosteroid signaling [54,55]. Contrasting regulation of these genes can be explained by studies in which BRs barley mutants showed a delayed wilting time and turgor under water deficiency compared to Bowman [56]. This property may be due to the semi-dwarf phenotype and a lower biomass of the above-ground parts of plants, which is associated with a lower demand for water in drought but it changes with the duration of stress. Similar observations were made for a dwarf *Arabidopsis* mutant, which constitutively uses less water under drought than the wild type but that doesn't mean that it is drought resistant [57]. We noticed also three genes with opposite change of expression induced by late drought between genotypes. Two genes (HORVU3Hr1G105420, HORVU7Hr1G051310) associated with pathogenesis-related proteins showed contrasting regulation status between genotypes exposed to late drought. They were annotated to glucan endo-1,3-beta-glucosidase GI, which is an important hydrolytic enzyme abundant in many plant species after infection by different types of pathogens [58]. The reason for the upregulation of pathogen-resistance genes in response to drought might be that plants are especially prone to drought-disease interactions [59], probably this phenomenon may exist in BW885. Furthermore, two DEGs, annotated as protein kinase (HORVU2Hr1G117610) and peroxidase (HORVU3Hr1G074950), were found, whose expression status changed from downregulation in control to upregulation at late drought. It would seem that such genes should be generally upregulated in response to stressor as indicated i.a. by Su et al. [60] who cloned wheat peroxidase (PRX) gene and observed that its expression was upregulated by drought, salt, $H_2O_2$, and ABA treatments.

## Differentially expressed genes annotated to phytohormones

In addition, the present study sought to understand the behavior of genes associated with main phytohormone classes affecting branching under drought. Firstly, Gene Ontology terms related to BRs metabolism and signaling were identified. This revealed four DEGs between drought and control in BW and all of them were downregulated at late time point of drought. In turn, gene HORVU4Hr1G032830 showed increased expression in BW885 under drought, regardless of time points. Moreover, all of them belonged to cytochrome P450 family. Interestingly, no significant expression changes in the *HvBRI1* (*uzu1.a*, HORVU3Hr1G068000) gene in crown tissue of BW and BW885 were observed in our experiment. In turn, analyzing GAs-related DEGs, we found that *uzu1.a* disfunction did not affect the expression of two crucial GAs biosynthetic genes, since HORVU3Hr1G022840 (*HvGA3ox2*) and HOR-VU5Hr1G124120 (*HvGA20ox1*) were down- and upregulated, respectively, in response to late drought in both genotypes. Otherwise, HORVU1Hr1G086810 (encoding GA2-oxidase) and

HORVU6Hr1G028790 (encoding TFs of WRKY family) were upregulated only in the barley *uzu1.a* mutant at late drought. In most studies, increased expression of genes encoding GA2-oxidase, causing a dwarf phenotype by decreasing internode elongation, was observed in plants exposed to abiotic stresses [61,62]. However, Mikołajczak et al. [10] noted that *GA2ox* DEGs identified in the same plant tissue (crown) like in the present study, were downregulated in response to heat stress. In our research only one DEGs (*HvHTD3*) associated with strigolactones was negatively induced by late drought in both genotypes. SLs directly participate in plant tolerance to abiotic stresses, but plants may adopt different survival strategies to cope with the water deficit, so genes associated with SLs may react ambiguously. For instance, the expression levels of SLs biosynthesis genes and SLs content decreased in tomato roots under drought stress [63]. By contrast, SLs levels were elevated in rice roots in response to water deficiency [64]. Any indication about behavior of SLs-related genes is of great importance, because they are still insufficiently investigated, especially in barley [65].

Noteworthy, we found two candidates to be linkers of BRs and GAs crosstalk, i.e., HORVU2Hr1G104040 and HORVU6Hr1G068980, both belonging to bHLH transcription factors family. The first one was upregulated and the second one reduced its expression in BW under drought. Plant basic helix–loop–helix (bHLH) transcription factors are widely known regulators of many biological processes, including response to hazardous agents [66]. It should be noted that the downregulated BRs biosynthetic and responsive genes, as well as lowered GAs signaling, contribute to the defects of plant growth [67], and that cv. Bowman is characterized by high stature phenotype. The analyzed mutant BW885 (*uzu1.a*) of semi-dwarf phenotype carries missense mutations in different domains of the BRs receptor encoded by the *HvBRI1* gene [34], being not differentially expressed in our study. However, we found a mutant-specific overexpression of another gene HORVU5Hr1G046390 annotated to WRKY domain which, according to KEGG/Gramene, is involved in signaling of brassinosteroid and jasmonic acid simultaneously. Indeed, Chen and Yin [68] reported that WRKY TFs were implicated in BRs signaling through interaction with BES1 TFs, and parallelly they modulated negatively the drought tolerance of *Arabidopsis* plants by massive repression of others drought-inducible gene. These findings suggested that overexpression of HORVU5Hr1G046390 may partially compensate the impairment of the important gene *HvBRI1* in the mutant. On the other hand, above-mentioned facts about WRKY functioning undermined to some extent the reports about utilization of this mutant in potential genetic building blocks for breeding strategies with sturdy and climate-tolerant barley cultivars.

## Differentially expressed genes annotated to drought and oxidative stress

We found several DEGs encoding dehydrins that are shared between genotypes in response to late drought. This is consistent with the report by Mikołajczak et al. [40], who also identified genes of various functions (including dehydrins) to be affected by drought and heat in barley flag leaf, regardless of the genotype. Such genes can underlie the universal stress response and, therefore, may be a promising target for barley improvement to better deal with changing climate. In contrast, there were two DEGs (HORVU1Hr1G016080 and HORVU4Hr1G074130) between genotypes belonging to annexin superfamily, with opposite regulation of expression inducted by late drought; in fact, they were downregulated in Bowman, but upregulated in BW885. There is some indications that enhanced drought tolerance could be due to annexin-mediated modulation of redox signaling network, which integrates with phytohormone-activated pathways [69]. Szalonek et al. [70] demonstrated that annexins are a promising target for manipulation of plant tolerance to stress stimuli. Regarding the analyzed mutant BW885, the annexin encoding gene overexpression may confer the greater tolerance to drought stress.

Moreover, 13 DEGs related to peroxidase showed greater expression in BW885 than in BW at late drought, and none in earlier drought. Peroxidases are enzymes involved in plant development and defense mechanisms against stress factors [71]. Our results indicated that BW885 may have different sensitivity to the drought signals than Bowman, thus influencing its overall drought response, unlike BW which did not need to initiate expression changes to a such degree when the stress occurred. Curiously, Janiak et al. [72] observed that the drought sensitive barley cultivar exhibited far more expression changes in response to drought stress than the tolerant genotype.

## Differentially abundant proteins and comparison of protein abundance vs. gene expression

We performed the global transcriptomic and proteomic profiling of barley crowns that were sampled simultaneously in the same experiment, which is not a usual approach. Mostly, such studies are separately performed. Interestingly, the comparison of protein and gene expression revealed that, despite the smaller overall number of differentially accumulated proteins, their general stress-induced behavior was similar to DEGs. It is worth to notice that, at the late time point, there were incredibly more proteins (the same as for genes) specific to Bowman, which means that large set of genes and proteins were developmentally-specific in control versus drought. A similar trend was observed for BW885, but they were less numerous. Nonetheless, the significant relationship between changes at transcriptomic and proteomic levels was identified only in Bowman under late stress (47 cases). Herein, the group of proteins involved in drought response (LEA proteins, chaperones HSP20 or AAI domain containing protein) had increased expression along with corresponding genes. This is not surprising since protective role of LEA proteins during various stresses has been well documented [73]. In turn, heat shock proteins, initially suggested to be involved in temperature stress response, have also been recently implicated in the response to other abiotic stresses [74]. Genes containing an AAI domain form a large gene family, but there have been no comprehensive analyses of this gene family in any plant species [75]. KEGG pathway annotation showed that the AAI domain containing protein was involved in the regulation of leaf development, which seems to be an important aspect for plant performance under constrained environment but requires further investigation. Therefore, converging the above-mentioned gene and protein expression can be of special interest and an extremely valuable approach for cereals' improvement in the future.

Among genes and linked proteins that were mutually downregulated under drought, we found HORVU7Hr1G095900 (annotated to cytochrome P450) with assigned unnamed protein being involved in tricin biosynthesis. Tricin is a specialized metabolite belonging to flavonoids and is involved in lignification, thus also conferring stress tolerance [76]. There were also two genes whose proteins were associated with the NAC5 transcription network. These transcription factors affect the expression of numerous genes related to stress tolerance, especially to drought and high salinity. On the other hand, they regulate many aspects of plant development, from juvenile growth to ripening [77]. Perhaps reduced accumulation of these proteins in BW resulted from the demand of limitation of growth and green biomass production under prolonged, severe drought. We found also four cases of reverse regulation of the gene and protein. Apparently, some post-transcriptional and/or post-translational modifications occurred in these cases, or stress treatment desynchronized the balance between transcriptional and translational regulation. For instance, Bowman-Birk inhibitor (BBI) was downregulated in BW at 10D, but the parallelly corresponding gene showed increased expression. BBIs belong to large and well documented protease inhibitor family found in many plant

species and are considered as members of plant defense system. It has been suggested that increased accumulation of BBIs may alleviate the drought-induced oxidative stress [78].

Overall impression might be confusing, since one mechanism led to better resistance of Bowman or BW885 to drought, but others acted in the opposite way; however, the response to drought is a very complex phenomenon and many factors may show multiple functions, such as plant growth regulators as well as protective agents. In fact, a proper balance is required in plant to survive under adverse conditions.

## Phytohormones, sterols and phenotypic characterization

Finally, we enriched our research with three main classes of brassinosteroids: brassinolide and its biosynthetically related precursors (castasterone and cathasterone). Brassinolide was the dominant brassinosteroid in the crown tissue, especially in Bowman. Generally, castasterone is suggested to show the highest biological activity among BRs hormones in monocots, including barley [56,79]; however, this finding is primarily ground on the leaves-based study. Apparently, in the crown, brassinolide may act more stimulatively than castasterone, thus facilitating the development of lateral shoots. Consequently, the higher concentration of brassinolide in the crown, where the branching is controlled, may lead to lower levels of castasterone, as brassinolide promotes the branching of shoots at the expense of castasterone activity; however, this aspect requires further investigation.

The correlation between BRs and abiotic stress response has been reported [80]; however, both positive and negative effects of BRs have been suggested, depending on their concentration, i.e. over-accumulation may be deleterious [27]. We proved that temporal drought induced the opposite hormonal reaction of genotypes with different BRs signaling efficiency. In BW, the level of brassinolide and castasterone increased as a result of prolonged drought compared to early stress, whereas in BW885 it decreased at the late time point compared to the early one. Our findings indicated that, among studied brassinosteroids, the rapid accumulation of castasterone by the *uzu1.a* mutant was the main determinant of its different performance under drought, of note it was accumulated 7-fold more in BW885 than in BW at early stress. In addition, it cannot be excluded that alternative pathways to BRI1 receptor signaling exists in the *uzu1.a* mutant, which may require a higher content of brassinolide precursors; one example could be BRs signaling mediated by G-protein α-subunit, which is an important player in plant signalosome including phytohormones pathways. In any case, further detailed investigation is required to verify this bold hypothesis. Moreover, other precursors in the biosynthesis of all BRs are sterols and we confirmed that sitosterol and campesterol were the major components of the sterol fraction in the crown tissue of both genotypes. Noteworthy, this is a pioneer report about sterols quantification in barley crowns. In general, drought resulted in increased accumulation of phytosterols, but in contrast, avenasterol decreased in all cases. Our previous study performed under drought, heat, and salinity, acting separately and simultaneously, demonstrated that avenasterol was not significantly affected by abiotic stress in leaves of barley genotypes without any perturbations in BRs metabolism [35]. This may suggest a unique role of avenasterol in plant's reaction to stress, but it has not been well recognized to date. Only Wang et al. [81] demonstrated that avenasterol exhibits antioxidant properties at high temperatures due to the presence of an ethylidene group in its side chain, which leads to the formation of an allylic free radical that can isomerize to other relatively stable free radicals [82]. We also found that cholesterol level, unlike others, increased during drought in barley mutant. Nevertheless, it is possible that the fluctuations in the sterol composition may be essential for certain processes related to plant growth and development, as well as processes involved in stress compensation [83].

Phenotyping confirmed the general phenotypic effect of the *HvBRI1* mutation. Overall, it was noticeable that the *uzu1.a* mutant was characterized by lower plant height and later heading than BW. Interestingly, reduced height of mutant resulted majorly from shortened peduncle, similarly to the gibberellin-deficient *sdw1* barley mutant, as demonstrated in our previous study [10]. The overall drought effect on genotypes was bigger for BW885 than for Bowman. This finding was surprising, since we discovered some molecular evidence suggesting better resistance of the *uzu1.a* mutant under drought, which has also been proposed by other researchers. Most likely, better yielding of the *uzu1.a* barley mutant is manifested especially under field conditions, where reduced lodging, a known property of semi-dwarf plants, may effectively increase the yield, as suggested by Chono et al. [22]. It must be stressed out that during plant growth in a pot in greenhouse experiment the lodging is generally a very limited phenomenon.

## Conclusions

In our study independent research methods, performed on barley crown tissue, allowed to identify genes and corresponding proteins, whose direction of regulation was the same under water deficit conditions. They may represent a promising set of genes in breeding of new varieties with increased drought tolerance. Using the *uzu1.a* mutant we clearly demonstrated that disturbances in brassinosteroids signal transduction affected the molecular response to drought. A set of genes with contrasting regulation between Bowman and the *uzu1.a* mutant was determined, often of fundamental role in plant development. Meanwhile, several genes encoding dehydrins were expressed independently from genotype, thus being suggested as central agents during drought response. Some candidate genes on coordination of phytohormones crosstalk have been also proposed. Moreover, transcriptome and proteome studies proved that changes caused by early and late drought were different, regardless of the genotype, i.e. independently from perturbations in BRs perception. Overall, the present study revealed the specific stress-induced re-modeling of the barley *uzu1.a* mutant's behavior at various levels, from transcriptomic through proteomic and phytohormonal to phenotypic. The multidisciplinary results provided new insights into the significant relationships between gene expression, protein, and phytohormone content of crown tissue under abiotic stress.

## Supporting information

**S1 Table. Optimized HPLC/ESI-MS/MS parameters in multiple reaction mode (MRM) with positive ionization are listed for the brassinosteroids quantified in barley.** The fragmentation parameters refer to the first Q3, which was used for quantification.
(DOCX)

**S2 Table. Identified SNPs with assigned genes and predicted protein translation effects.**
(XLSX)

**S3 Table. Results of differential expression analysis with gene annotation.**
(XLSB)

**S4 Table. GO term overrepresentation for sets of DEGs in treatment and genotype comparisons.**
(XLSX)

**S5 Table. GO terms used to select genes related to gibberellin, brassinosteroid, strigolactone, drought and oxidative stress.**
(XLSX)

**S6 Table. Differentially expressed genes related to brassinosteroid, gibberellin, strigolactone, drought and oxidative stress identified using GO annotation (1) or KEGG and Gramene databases (2).**
(XLSX)

**S7 Table. Differentially abundant proteins in D vs. C comparison for Bowman, BW885 at 3D and 10D, along with regulation status of assigned DEGs.**
(XLSX)

**S8 Table. Results of ANOVA for sterol classes (A); Phenotypic traits observed in the experiment with corresponding symbol (ID), units, description, ontology annotation identifier and ANOVA results (B).**
(XLSX)

**S1 Fig. Extracted ion chromatograms (XIC) of MRM and enhanced product ion (EPI) spectra of the brassinosteroids quantified in barley samples.** BL, brassinolide; CS, castasterone; CT, cathasterone.
(TIF)

**S2 Fig.** Mean values (with standard errors) of brassinosteroids content in Bowman and BW885 under control (C) and drought (D) measured one day before stress application (0D), in 3$^{rd}$ day (3D), and 10$^{th}$ day (10D) of stress.
(TIF)

**S3 Fig.** Mean values (with standard errors) of phytosterols content in Bowman and BW885 under control (C) and drought (D) measured one day before stress application (0D), in 3$^{rd}$ day (3D), and 10$^{th}$ day (10D) of stress.
(TIF)

**S4 Fig.** Mean values (with standard errors) of phenotypic traits observed for Bowman and BW885 under control (C) and drought (D); (B) principal component biplot for analyzed traits (T1-T22): green circle, BW_C; red circle, BW_D; green triangle, BW885_C; red triangle, BW885_D.
(PDF)

## Acknowledgments

Authors would like to thank Dr. Natalia Witaszak for performing the proteomic quantification and Dr. Alberto Adeva from the Scientific and Technological Centers of the University of Barcelona (CCiTUB) for his assistance in the brassinosteroids analyses. Computations were performed in part with the support of Poznań Supercomputing and Networking Centre (http://www.man.poznan.pl).

## Author Contributions

**Conceptualization:** Anetta Kuczyńska, Piotr Ogrodowicz, Krzysztof Mikołajczak.

**Data curation:** Paweł Krajewski.

**Formal analysis:** Anetta Kuczyńska, Paweł Krajewski, Krzysztof Mikołajczak.

**Funding acquisition:** Anetta Kuczyńska.

**Investigation:** Anetta Kuczyńska, Piotr Ogrodowicz, Michał Kempa, Vladimiro Cardenia, Maria Teresa Rodriguez-Estrada, Marina Pérez-Llorca, Krzysztof Mikołajczak.

**Methodology:** Anetta Kuczyńska, Piotr Ogrodowicz, Michał Kempa, Vladimiro Cardenia, Maria Teresa Rodriguez-Estrada, Marina Pérez-Llorca, Sergi Munné-Bosch, Krzysztof Mikołajczak.

**Project administration:** Anetta Kuczyńska.

**Resources:** Anetta Kuczyńska.

**Software:** Paweł Krajewski.

**Supervision:** Anetta Kuczyńska.

**Validation:** Anetta Kuczyńska, Paweł Krajewski, Krzysztof Mikołajczak.

**Visualization:** Anetta Kuczyńska, Martyna Michałek, Paweł Krajewski, Krzysztof Mikołajczak.

**Writing – original draft:** Anetta Kuczyńska, Paweł Krajewski, Vladimiro Cardenia, Maria Teresa Rodriguez-Estrada, Sergi Munné-Bosch, Krzysztof Mikołajczak.

**Writing – review & editing:** Anetta Kuczyńska, Martyna Michałek, Piotr Ogrodowicz, Michał Kempa, Paweł Krajewski, Vladimiro Cardenia, Maria Teresa Rodriguez-Estrada, Sergi Munné-Bosch, Krzysztof Mikołajczak.

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
