## [Decision Letter · Decision Letter 0]

12 Jul 2024

PONE-D-24-21000Disorders in brassinosteroids signal transduction triggers the profound molecular alterations in the crown tissue of barley under droughtPLOS ONE

Dear Dr. Mikołajczak,

Thank you for submitting your manuscript to PLOS ONE. After careful consideration, we feel that it has merit but does not fully meet PLOS ONE’s publication criteria as it currently stands. Therefore, we invite you to submit a revised version of the manuscript that addresses the points raised during the review process.

We look forward to receiving your revised manuscript.

Kind regards,

Shailender Kumar Verma, Ph.D.

Academic Editor

PLOS ONE

 [This work was supported by the National Science Centre, Poland, project Opus 12 no. 2016/23/B/NZ9/03548.].  

Reviewers' comments:

Reviewer's Responses to Questions

**Comments to the Author**

1. Is the manuscript technically sound, and do the data support the conclusions?

Reviewer #1: Partly

Reviewer #2: Partly

2. Has the statistical analysis been performed appropriately and rigorously? 

Reviewer #1: I Don't Know

Reviewer #2: Yes

3. Have the authors made all data underlying the findings in their manuscript fully available?

Reviewer #1: Yes

Reviewer #2: No

4. Is the manuscript presented in an intelligible fashion and written in standard English?

Reviewer #1: No

Reviewer #2: No

5. Review Comments to the Author

Reviewer #1: In “Disorders in brassinosteroids signal transduction triggers the profound molecular alterations in the crown tissue of barley under drought”, the authors describe a study examining the genomes, transcriptomes, and proteomes, and agronomic traits of two contrasting lines of barley, one of which encodes a mutation in the brassinosteroid receptor BRI1 (uzu1.a mutant). The manuscript includes extensive exploration of genes showing differential expression due to drought or genotype. Generally the study addresses understanding the roles of hormones in the major crop barley is valuable research. The bioinformatics for the transcriptome are appropriate. The supplementary data tables are transparent for evaluating the findings.

Major comments

I was unable to identify the figure legends in the submission.

The motivation/relevance/justification for several of the experiments is difficult to identify, especially in the introduction and results sections. Most importantly, the authors examine the crown tissue of barley in the study. However, they do not provide information about whether or how brassinosteroids affect the crown (or tillering) in barley (or even other species) to motivate the omics approaches on this choice of tissue. As such, the examination of tillering genes was also poorly motivated in the results or introduction. Restructuring and more detail added to the introduction and results will likely address this.

Further, the genomic comparison between lines requires more motivation—especially because the study was primarily transcriptomic and discussion of promoter SNPs was not discussed. Further, whether the SNPs are located in the regions that is supposed to be diverging between the NILs is not described. Providing

Minor comments

The conditions T1 and T2 have the same names as the agronomic traits measures (T1-T26). Please rename T1 and T2 to “3D” and “10D” to disambiguate the overlap and make it easier for the reader to follow the paper.

Reviewer #2: Dear Authors,

In this manuscript (Disorders in brassinosteroids signal transduction triggers the profound molecular alterations in the crown tissue of barley under drought) authors investigated how the response to drought stress in barley crown tissue occurs at the molecular level and were interested in how this response is primarily influenced by the brassinosteroid signalling pathway. The manuscript addresses a topical issue and is of interest to the scientific community, but unfortunately I cannot recommend it for publication in this form, as it contains flaws mainly in the methodological part.

The major comments are listed below:

1.Keywords- do not correctly capture the meaning of the manuscript, should be corrected.

2.The section "Determination of phytohormone and sterol composition" should be rewritten and supplemented with relevant information. The citation of Dias et al. 2018, which is supposed to describe the LC-MS/MS methodology for the quantification of BRs, is not listed in the reference list, nor have I been able to locate such an article. So this methodology should be described in detail!

3. I assume that if 100 mg of plant material was used, this extract must have been purified somehow to remove ballast substances? Please provide details.

4. It is known from the available literature that 6-keto brassinosteroids (CS, 28-homoCS, norCS) occur in higher concentrations in cereals (wheat, barley), so it is interesting that you have quantified BL at relatively high levels (40-70 ng/g), this should be adequately discussed.

Minor points:

1.There are a number of errors in citations throughout the text and many of the abbreviations used are not explained; this should be checked carefully.

2. The “Discussion” section is very long and the reader finds it difficult to navigate. In my opinion, it should be shortened.

6. PLOS authors have the option to publish the peer review history of their article (what does this mean?). If published, this will include your full peer review and any attached files.

Reviewer #1: No

Reviewer #2: No

---

## [Author Response · Author response to Decision Letter 0]

12 Sep 2024

We would like to thank you for the suggestions and comments, which have been useful for improving the quality of the manuscript entitled “Disorders in brassinosteroids signal transduction triggers the profound molecular alterations in the crown tissue of barley under drought” by Anetta Kuczyńska, Martyna Michałek, Piotr Ogrodowicz, Michał Kempa, Paweł Krajewski, Vladimiro Cardenia, Maria Teresa Rodriguez-Estrada, Marina Pérez-Llorca, Sergi Munné-Bosch and Krzysztof Mikołajczak. 

An itemized list has been prepared for the Reviewers, stating how each point was addressed and modified according to the suggestions. We believe that all objections were properly faced and a suitable answer/modification was provided.

Reviewer #1: In “Disorders in brassinosteroids signal transduction triggers the profound molecular alterations in the crown tissue of barley under drought”, the authors describe a study examining the genomes, transcriptomes, and proteomes, and agronomic traits of two contrasting lines of barley, one of which encodes a mutation in the brassinosteroid receptor BRI1 (uzu1.a mutant). The manuscript includes extensive exploration of genes showing differential expression due to drought or genotype. Generally the study addresses understanding the roles of hormones in the major crop barley is valuable research. The bioinformatics for the transcriptome are appropriate. The supplementary data tables are transparent for evaluating the findings.

Major comments

I was unable to identify the figure legends in the submission.

Figure captions have been placed directly after the paragraph in which they are first cited in the text, according to the journal's guidelines. According to the reviewer's suggestion, the figure legend has also appeared before the Supporting Information.

The motivation/relevance/justification for several of the experiments is difficult to identify, especially in the introduction and results sections. Most importantly, the authors examine the crown tissue of barley in the study. However, they do not provide information about whether or how brassinosteroids affect the crown (or tillering) in barley (or even other species) to motivate the omics approaches on this choice of tissue. As such, the examination of tillering genes was also poorly motivated in the results or introduction. Restructuring and more detail added to the introduction and results will likely address this.

We thank the Reviewer for the suggestion. The examination of the main tillering genes was removed from the manuscript, because their behavior did not change significantly in our study. Moreover, this also allowed the reduction of the paper length as requested by Reviewer 2.

We have also enriched the manuscript with additional information related to connection of brassinosteroid and tillering (lines 121-132 in the Revised Manuscript with Track Changes):

It appears that since BRs promote cell division, a higher level of BRs in the crown should facilitate tillering. It has been established that the overexpression of miR397 activates the BR response and leads to an increase in tillering in rice [29]. It was also found that BRs promotes bud outgrowth in tomato, and BR signaling integrates multiple pathways that control shoot branching [30]. Noteworthy, the growth-promoting phytohormones BRs have distinct roles in regulating tillering in rice and rosette branching in Arabidopsis, respectively. In Arabidopsis, BRs do not regulate the primary branch number [31]; however, in rice, BRs significantly promote tillering [32]. Given that the relationship between BRs and tillering is understood, it should be noted that the functioning of BRs in the crown has not been previously investigated, so multiomic studies were designed in the crown of genotypes differentiated by their efficiency in BR signaling mediated by the BRI1 receptor.

Further, the genomic comparison between lines requires more motivation—especially because the study was primarily transcriptomic and discussion of promoter SNPs was not discussed. Further, whether the SNPs are located in the regions that is supposed to be diverging between the NILs is not described. Providing

As the genetic background difference for the near isogenic lines are clear we used SNP genotyping to identify the range of the polymorphism between Bowman and BW885 line. We proved that they were genetically more polymorphic than it has been commonly suggested so far. Thus, to reduce the influence of multiple introgressions, defined as regions containing SNPs, on gene expression analysis the polymorphic DEGs were filtered out from the biological interpretation; only those located in the introgressed region around the uzu1.a locus remained. Overall, such approach did not affect substantially our interpretation of differential gene expression analysis, since only about 5% of all DEGs contained SNPs (most of them were not of HIGH translation effect according to VEP tool). In a consequence, only few genes were removed from the discussion, namely, 1 DEG related to brassinosteroids, 1 DEG of ABC transporter, 1 DEG of oxidative stress); however, polymorphic DEGs are available in the supplementary data as the obtained result. Meanwhile, it should be bear in mind that some removed DEGs can play important regulatory functions – this also has been added to the text.

The relevant fragments were added to the manuscript:

- „…including 136 polymorphic genes between genotypes.” (lines 307-308 in the Revised Manuscript with Track Changes)

- “Thus, to reduce the influence of multiple introgressions, defined as regions containing SNPs, on gene expression analysis the polymorphic DEGs were filtered out from the biological interpretation; only those located in the introgressed region around the uzu1.a locus remained. This region was extrapolated roughly based on Dockter et al. study using the HvBRI1 (uzu1.a) gene position (464,679,001-464,682,900; Ensembl Plants), namely, it was the interval 460,879,001-467,372,900 of chromosome 3H (Ensembl Plants); in total seven DEGs were identified there. Nonetheless, such approach did not substantially affect our interpretation of differential gene expression analysis, since only about 5% of all DEGs contained SNPs. Meanwhile, it should be bear in mind that some removed DEGs can play important regulatory functions. For instance, two DEGs containing SNPs in the “upstream” region of the gene (in promoter sequence putatively) were annotated to heat shock factors (HSFs) regulatory network; in fact, they are suggested to participate in plant growth and stress response [51].” (lines 684 – 696 in Revised Manuscript with Track Changes) 

The small number of polymorphic DEGs between genotypes, 136 (ca. 5% of the total), prompted us to remove them from the interpretation of the results, which is now described in the text. This approach means that we do not further consider the effects of other polymorphic regions between genotypes including the analysis of promoter polymorphism. Nevertheless, on the suggestion of the reviewer, we isolated those genes (17) that had a homozygous SNP in the "upstream region" i.e. potentially related to the gene promoter, and only four of them were DEGs. This convinced us that elimination of polymorphic DEGs from the results interpretation was the suitable approach.

The relevant fragment was added to the manuscript;

“There were 22 homozygous SNPs (50k Chip) located in ‘upstream gene variant’ of 17 genes with ‘MODIFIER’ effect (Supplementary Table S1).” (lines 282-283 in the Revised Manuscript with Track Changes)

The consideration about the relationship between SNPs polymorphism and differential gene expression analysis was removed; therefore, Results and Discussion sections were shortened accordingly.

Minor comments

The conditions T1 and T2 have the same names as the agronomic traits measures (T1-T26). Please rename T1 and T2 to “3D” and “10D” to disambiguate the overlap and make it easier for the reader to follow the paper.

As suggested by the Reviewer we have changed throughout the publication T1 -> 3D and T2 -> 10D.

Reviewer #2: Dear Authors,

In this manuscript (Disorders in brassinosteroids signal transduction triggers the profound molecular alterations in the crown tissue of barley under drought) authors investigated how the response to drought stress in barley crown tissue occurs at the molecular level and were interested in how this response is primarily influenced by the brassinosteroid signalling pathway. The manuscript addresses a topical issue and is of interest to the scientific community, but unfortunately I cannot recommend it for publication in this form, as it contains flaws mainly in the methodological part.

The major comments are listed below:

1.Keywords- do not correctly capture the meaning of the manuscript, should be corrected.

Thank you for this suggestion. Keywords have been changed as follows:

abiotic stress, crown, functional annotation, Hordeum vulgare L., phytohormones, stress-induced proteins, transcriptomics, uzu1.a

2.The section "Determination of phytohormone and sterol composition" should be rewritten and supplemented with relevant information. The citation of Dias et al. 2018, which is supposed to describe the LC-MS/MS methodology for the quantification of BRs, is not listed in the reference list, nor have I been able to locate such an article. So this methodology should be described in detail!

We thank the Reviewer for bringing this to our attention. According to the suggestion, the citation of Dias et al. 2018 has been added and the section “Determination of phytohormone and sterol composition” has been modified as follows (lines 233-239 in the Revised Manuscript with Track Changes): 

Briefly, extracts were obtained by vortexing, ultrasonication for 30 min and centrifugation (9500 g, 15 min, 4°C) of the mixture of plant material powder and extraction solvent (methanol and 1% glacial acetic acid) containing labelled standards (d3 – BL, d3 – CS, d3 – CT). The extracts were then filtered with 0.22 µm PTFE filters (Phenomenex, Torrance, CA, USA) prior to injection into the UHPLC-MS/MS system. The measurements were conducted by multiple reaction monitoring (MRM) in positive ion mode; technical details are reported in Setsungnern et al. [43].

3. I assume that if 100 mg of plant material was used, this extract must have been purified somehow to remove ballast substances? Please provide details.

The missing details have been placed in the lines 236-238 in the Revised Manuscript with Track Changes: 

The extracts were then filtered with 0.22 µm PTFE filters (Phenomenex, Torrance, CA, USA) prior to injection into the UHPLC-MS/MS system.

4. It is known from the available literature that 6-keto brassinosteroids (CS, 28-homoCS, norCS) occur in higher concentrations in cereals (wheat, barley), so it is interesting that you have quantified BL at relatively high levels (40-70 ng/g), this should be adequately discussed.

As suggested by the Reviewer the high level of brassinolide obtained in the study has been discussed (lines 950-956 in the Revised Manuscript with Track Changes): 

Generally, castasterone is suggested to show the highest biological activity among BRs hormones in monocots including barley [57, 80]; however, this finding is primarily ground on the leaves-based study. Apparently, in the crown, brassinolide may act more stimulatively than castasterone, thus facilitating the development of lateral shoots. Consequently, the higher concentration of brassinolide in the crown, where the branching is controlled, may lead to lower levels of castasterone, as brassinolide promotes the branching of shoots at the expense of castasterone activity; however, this aspect requires further investigation.

Minor points:

1.There are a number of errors in citations throughout the text and many of the abbreviations used are not explained; this should be checked carefully.

We thank the Reviewer for the suggestion. As recommended, we have carefully checked the entire text and the citations are corrected and the abbreviations are explained.

2. The “Discussion” section is very long and the reader finds it difficult to navigate. In my opinion, it should be shortened.

The discussion has been shortened based on the results along with changes suggested by the Reviewers. 

The consideration about the relationship between SNPs polymorphism and differential gene expression analysis was removed; therefore, Results and Discussion sections were shortened accordingly, and three tables were deleted. Also, some fragments concerning less interesting observations were deleted to reduce the length of the manuscript (e.g. lines 697-713, 732-750, 763-773, 848-858, 883-896 in the Revised Manuscript with Track Changes). Additionally, to facilitate the readability and navigation through the discussion few subheadings were introduced into the Discussion section.

---

## [Decision Letter · Decision Letter 1]

29 Oct 2024

PONE-D-24-21000R1Disorders in brassinosteroids signal transduction triggers the profound molecular alterations in the crown tissue of barley under droughtPLOS ONE

Dear Dr. Mikołajczak,

Thank you for submitting your manuscript to PLOS ONE. After careful consideration, we feel that it has merit but does not fully meet PLOS ONE’s publication criteria as it currently stands. Therefore, we invite you to submit a revised version of the manuscript that addresses the points raised during the review process.

We look forward to receiving your revised manuscript.

Kind regards,

Shailender Kumar Verma, Ph.D.

Academic Editor

PLOS ONE

Reviewers' comments:

Reviewer's Responses to Questions

**Comments to the Author**

1. If the authors have adequately addressed your comments raised in a previous round of review and you feel that this manuscript is now acceptable for publication, you may indicate that here to bypass the “Comments to the Author” section, enter your conflict of interest statement in the “Confidential to Editor” section, and submit your "Accept" recommendation.

Reviewer #2: (No Response)

2. Is the manuscript technically sound, and do the data support the conclusions?

Reviewer #2: No

3. Has the statistical analysis been performed appropriately and rigorously? 

Reviewer #2: Yes

4. Have the authors made all data underlying the findings in their manuscript fully available?

Reviewer #2: No

5. Is the manuscript presented in an intelligible fashion and written in standard English?

Reviewer #2: Yes

6. Review Comments to the Author

Reviewer #2: Dear authors,

Dias et al. 2018 and Setsungnern et al. 2020 do not provide validation parameters for the UHPLC-MS/MS method; if this method has been properly validated, please provide the appropriate citation where these data can be found. If not, I strongly recommend including the corresponding chromatograms and spectral data in the Supplements.

Did you made sure that matrix effects, when working with 100 mg of plant material without using SPE or derivatization , is not disturbing your quantification?

7. PLOS authors have the option to publish the peer review history of their article (what does this mean?). If published, this will include your full peer review and any attached files.

Reviewer #2: No

---

## [Author Response · Author response to Decision Letter 1]

15 Nov 2024

Dear Reviewer,

To address your comment regarding the methodology for phytohormone determination, we have significantly expanded the description in the Materials and Methods section (lines 228-242). Additionally, we have introduced a supplementary table (Supplementary Table S1) presenting the optimized MRM parameters, and, as per your suggestion, we have included a supplementary figure (Figure S1) featuring the extracted ion chromatograms (XIC) of MRM and enhanced product ion (EPI) spectra of the brassinosteroids analyzed in the barley samples. These revisions have resulted in a renumbering of the supplementary materials. The citation Dias et al. 2018 has been removed that resulted in renumbering of cited literature within the manuscript. 

Did you made sure that matrix effects, when working with 100 mg of plant material without using SPE or derivatization, is not disturbing your quantification?

Response: Matrix effects are certainly reducing a little bit the sensitivity of the MS/MS but it is not disturbing the quantification. Peaks were in some cases small but clear enough to allow a proper quantification (see newly submitted suppl. file). 

Many thanks for all your comments and criticisms raised, which have served to improve our work. We hope these changes comprehensively address your concerns and enhance the clarity and robustness of the manuscript.

---

## [Decision Letter · Decision Letter 2]

30 Dec 2024

PONE-D-24-21000R2Disorders in brassinosteroids signal transduction triggers the profound molecular alterations in the crown tissue of barley under droughtPLOS ONE

Dear Dr. Mikołajczak,

Thank you for submitting your manuscript to PLOS ONE. After careful consideration, we feel that it has merit but does not fully meet PLOS ONE’s publication criteria as it currently stands. Therefore, we invite you to submit a revised version of the manuscript that addresses the points raised during the review process.

We look forward to receiving your revised manuscript.

Kind regards,

Nguyen Hoai Nguyen

Academic Editor

PLOS ONE

**Journal Requirements:**

Reviewers' comments:

Reviewer's Responses to Questions

**Comments to the Author**

1. If the authors have adequately addressed your comments raised in a previous round of review and you feel that this manuscript is now acceptable for publication, you may indicate that here to bypass the “Comments to the Author” section, enter your conflict of interest statement in the “Confidential to Editor” section, and submit your "Accept" recommendation.

Reviewer #2: All comments have been addressed

2. Is the manuscript technically sound, and do the data support the conclusions?

Reviewer #2: (No Response)

3. Has the statistical analysis been performed appropriately and rigorously? 

Reviewer #2: (No Response)

4. Have the authors made all data underlying the findings in their manuscript fully available?

Reviewer #2: (No Response)

5. Is the manuscript presented in an intelligible fashion and written in standard English?

Reviewer #2: (No Response)

6. Review Comments to the Author

**Reviewer #2: **The authors have added the required information regarding LC-MS analysis.I agree to accept this manuscript.

7. PLOS authors have the option to publish the peer review history of their article (what does this mean?). If published, this will include your full peer review and any attached files.

Reviewer #2: No

---

## [Author Response · Author response to Decision Letter 2]

12 Jan 2025

We would like to thank you for the suggestions and comments, which have been useful for improving the quality of the manuscript entitled “Disorders in brassinosteroids signal transduction triggers the profound molecular alterations in the crown tissue of barley under drought” by Anetta Kuczyńska, Martyna Michałek, Piotr Ogrodowicz, Michał Kempa, Paweł Krajewski, Vladimiro Cardenia, Maria Teresa Rodriguez-Estrada, Marina Pérez-Llorca, Sergi Munné-Bosch, and Krzysztof Mikołajczak. 

Noteworthy, in this round of the revision no additional reviewers’ comments required addressing. However, the Editor suggested via email message (the original message bellow) to double check the manuscript to make sure there are no typos, which has been carefully done. The file Manuscript with Tracking Changes.doc reflects these corrections. Additionally, the funding information has been removed from the main text, as requested by the editorial office.

We hope these changes comprehensively addresses Editor’s concerns.

“Dear Dr. Krzysztof Mikołajczak,

Please double check your manuscript once again to make sure there are no typos in the manuscript. Please note that the microRNA names should be italicized as well. After that, please resubmit the manuscript via the PLOS ONE system. 

Thank you very much.

Best regards,

Nguyen”

---

## [Editor Report · Decision Letter 3]

14 Jan 2025

Disorders in brassinosteroids signal transduction triggers the profound molecular alterations in the crown tissue of barley under drought

PONE-D-24-21000R3

Dear Dr. Mikołajczak,

We’re pleased to inform you that your manuscript has been judged scientifically suitable for publication and will be formally accepted for publication once it meets all outstanding technical requirements.

Kind regards,

Nguyen Hoai Nguyen

Academic Editor

PLOS ONE
---

## [Editor Report · Acceptance letter]

17 Jan 2025

PONE-D-24-21000R3 

PLOS ONE

Dear Dr. Mikołajczak, 

I'm pleased to inform you that your manuscript has been deemed suitable for publication in PLOS ONE. Congratulations! Your manuscript is now being handed over to our production team.

Kind regards, 

on behalf of

Dr. Nguyen Hoai Nguyen 

Academic Editor

PLOS ONE